# Current-induced domain wall motion in a van der Waals ferromagnet Fe₃GeTe₂

Wenjie Zhang[1,4], Tianping Ma [1,2,4] ✉, Binoy Krishna Hazra [1], Holger Meyerheim [1], Prajwal Rigvedi[1], Zihan Yin[1,3], Abhay Kant Srivastava[1], Zhong Wang [1,3], Ke Gu [1,3], Shiming Zhou[2], Shouguo Wang [2], See-Hun Yang[1], Yicheng Guan[1] & Stuart S. P. Parkin [1,3] ✉

The manipulation of spin textures by spin currents is of fundamental and technological interest. A particularly interesting system is the 2D van der Waals ferromagnet Fe₃GeTe₂, in which Néel-type skyrmions have recently been observed. The origin of these chiral spin textures is of considerable interest. Recently, it was proposed that these derive from defects in the structure that lower the symmetry and allow for a bulk vector Dzyaloshinsky-Moriya interaction. Here, we demonstrate current-induced domain wall motion in Fe₃GeTe₂ flakes, in which the maximum domain wall velocity is an order of magnitude higher than those reported in previous studies. In heterostructures with Pt or W layers on top of the Fe₃GeTe₂ flakes, domain walls can be moved via a combination of spin transfer and spin-orbit torques. The competition between these torques leads to a change in the direction of domain wall motion with increasing magnitude of the injected current.

The conversion of electric current to spin current plays a pivotal role in spintronics[1]. Spin currents that carry spin (and possibly orbital) angular momentum can be used to generate torques to manipulate local magnetizations. Such torques, whether spin transfer[2,3] or spin-orbit[4–6] derived, form the basic building blocks for the realization of novel spintronic devices. Among them, Racetrack memory devices based on the current-induced domain wall motion (CIDWM) driven by spin-transfer torques (STTs) and/or spin-orbit torques (SOTs) in magnetic nanowires are a leading candidate for next generation, non-volatile, memory devices with high speed, high density, and low energy consumption[1,7–9].

Magnetic two-dimensional van der Waals materials provide an emerging platform for spintronics research[10–12]. Among these materials, Fe₃GeTe₂ (FGT) stands out due to its metallic nature, tunable Curie temperature[13,14] and strong perpendicular magnetic anisotropy[15–18]. Recently, various chiral magnetic nanostructures have been observed in pristine FGT and heterostructures formed from it[19–23]. These chiral spin textures require a source of DMI. One source that was proposed was an interfacial DMI that arose from surface oxidation[21]. However,

given the substantial thickness of the FGT flakes in which the spin textures were observed this seems unlikely. Rather the origin should be bulk-like. Indeed, it was recently unambiguously found that FGT, both in bulk crystals and, more recently, in films prepared by molecular beam epitaxy (MBE), displays a crystal structure that has broken inversion symmetry[19,24]. The origin lies in differences in occupation of certain Fe Wykoff crystal sites. This significant finding is a convincing evidence for a bulk-like DMI[24]. We note that, recently, similar findings of broken inversion symmetry that leads to chiral spin textures have been found for the compound PtMnGa[25] and the 2D van der Waals material CrTe₂[26].

There has been some initial exploration of the current-induced manipulation of the magnetization of FGT[27–33]. Many of these studies use the anomalous Hall resistance to indirectly explore such switching[27,28,30,31,34]. Direct imaging via Lorentz transmission electron microscopy (LTEM), however, is limited to relatively thick FGT lamella that provide large enough signals[32,33]. Moreover, the velocity of the CIDWM was very low. Another powerful technique to image magnetic spin textures is magneto-optical Kerr (MOKE) microscopy. Indeed, this

[1]Max Planck Institute of Microstructure Physics, Halle (Saale) D-06120, Germany. [2]Anhui Key Laboratory of Magnetic Functional Materials and Devices, School of Materials Science and Engineering, Anhui University, Hefei 230601, China. [3]Institute of Physics, Martin Luther University, Halle-Wittenberg, Halle (Saale) D-06120, Germany. [4]These authors contributed equally: Wenjie Zhang, Tianping Ma. ✉e-mail: tpma@ahu.edu.cn; stuart.parkin@mpi-halle.mpg.de

technique was used to uncover two of the very first van der Waals ferromagnets, namely $Cr_2Ge_2Te_6$[10] and $CrI_3$[11]. Furthermore, Kerr microscopy has been used to detect the magnetization of thin FGT nanoflakes with high spatial resolution and strong contrast[35]. Here we use Kerr microscopy to explore CIDWM in heterostructures of FGT, with and without heavy metal platinum (Pt) or tungsten (W) layers. We show that the DWs are driven by STT alone or by a combination of STT and SOT. In the latter case, we find that the direction of the DW motion reflects a competition between STT and SOT mechanisms, allowing for a change in direction with increasing current density. Moreover, we find that the speed of the DWs is an order of magnitude higher than those previously reported[36], although still significantly lower than those found in racetracks formed from conventional magnetic thin film heterostructures.

## Results

Kerr microscopy was used to study the CIDWM in racetrack devices 15-µm long and 2–4-µm wide that were fabricated from pristine FGT flakes using methods similar to those used previously for conventional racetrack memory devices[8]. Exemplary Kerr images are shown in Fig. 1a: each micrograph shows an image of the racetrack after applying a sequence of 50 current pulses with a current density of $0.71 \times 10^{11}$ A m$^{-2}$ and a pulse width $\tau$ of 10 ns to the device at $T = 140$ K. The DW moves in the direction opposite to the current injection direction, which is a typical feature of STT-driven DW motion, in which the conduction electrons are majority spin-polarized[7]. It has been

reported that there also exists a so-called intrinsic SOT in pristine FGT systems without any HM layer[37,38], but the origin of such an SOT and how it will influence the magnetic dynamics such as CIDWM needs further investigation.

In our studies the velocity of the DW $v$ is determined as follows: a sequence of images of the static DW is taken in the Kerr microscope where one or a series of current pulses, each 5 to 80 ns long, is applied between each Kerr image. The position of the DW in each Kerr image is then extracted and plotted versus the corresponding accumulated duration time $t$ of the applied current pulses. $v$ is determined by a linear fit of the DW position versus $t$. A typical set of data for a 10.5 nm thick FGT flake at $T = 20$ K is shown in Fig. 1b, from which it is found that $v = 5.68$ m s$^{-1}$ at a current density $J = 2.41 \times 10^{11}$ A m$^{-2}$. $v$ as a function of $J$ at $T = 20$ K is shown in Fig. 1c. It is worth noting that there is an upper threshold current density $J_{th}^U \sim 2.5 \times 10^{11}$ A m$^{-2}$, above which the current heats up the FGT device beyond its Curie temperature (~150 K) through Joule heating, resulting in the nucleation of multi-domains instead of DW displacement. There is also a lower threshold current density $J_{th}^L \sim 1.8 \times 10^{11}$ A m$^{-2}$, below which there is no substantial DW motion. For the STT mechanism, there can be an intrinsic threshold current that depends on the competition between the field-like and damping-like STT torques[39,40]. In previous work on the CIDWM in FGT, the threshold current was stated to have an intrinsic origin[33] but extrinsic pinning from defects is also possible. For current densities between these two thresholds, we found a maximum velocity for the DWs of $v_{max} = 5.68$ m s$^{-1}$, which is one order of magnitude higher than

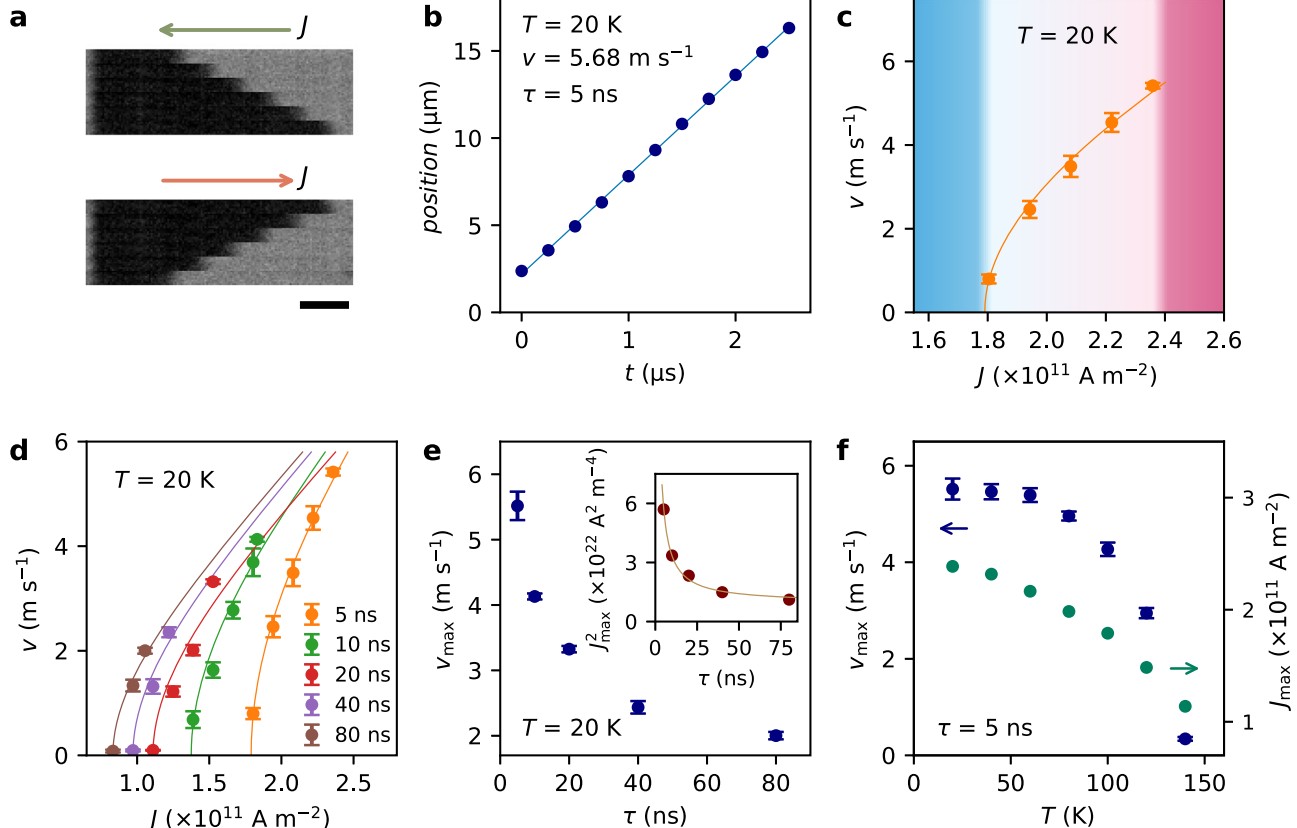

**Fig. 1 | CIDWM in FGT driven by STT. a** Differential Kerr microscope images of a single DW moving back and forth along a FGT device. Each image (from top to bottom) is taken after applying 50 current pulses ($\tau = 10$ ns) at $T = 140$ K. Green and orange arrows show the direction of the current. The scale bar corresponds to 2 µm. **b** DW velocity at $T = 20$ K found by fitting the DW position versus the accumulated time of the applied current pulses ($n \times 50 \times \tau$, $\tau = 5$ ns). **c** DW velocity as a function of current density $J$ at $T = 20$ K. There is no motion when $J$ is below a lower threshold value (blue shaded region). The DWs are eliminated due to Joule heating

when $J$ is above the upper threshold value (pink shaded region). Between these thresholds, the velocity increases as the current density increases, according to $v \propto \sqrt{J^2 - \left(J_{th}^L\right)^2}$. **d** DW velocity versus $J$ for different $\tau$ at $T = 20$ K. Solid lines are fits of $v$ vs $J$. **e** $v_{max}$ vs $\tau$ at $T = 20$ K. The inset shows a fit to the data. **f** $v_{max}$ (green circles) and the corresponding $J$ (navy squares) as a function of temperature for $\tau = 5$ ns. Error bars are determined by multiplying the standard error of the mean by a factor of 1.96 to establish a 95% confidence interval.

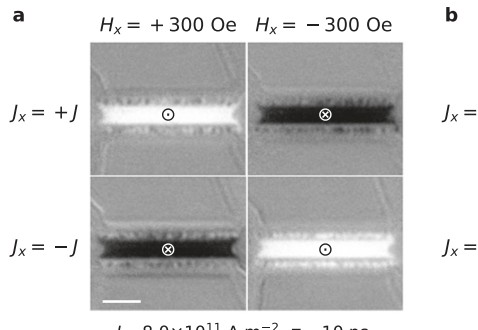

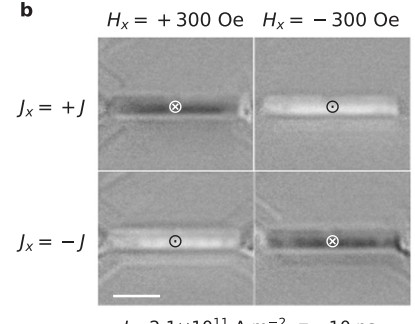

**Fig. 2 | Current-induced magnetization switching in FGT/Pt and FGT/W heterostructures.** Differential Kerr microscopy images of FGT(4.1 nm)/Pt(3 nm) (**a**) and FGT(8.6 nm)/W(3 nm) (**b**), where ⊙ and ⊗ represent magnetization up and down, respectively. 50 current pulses with $\tau = 10$ ns area applied to the heterostructures along the x axis under the application of a longitudinal field $H_x = 300$ Oe. The scale bars correspond to 5 µm.

those reported in previous work[33]. Nevertheless, higher $v$ could not be achieved due to Joule heating from the current that eventually heats up the sample beyond its Curie temperature ($T_c$), thereby erasing the domain walls. The relatively low maximum $v$ is due to the low $T_c$ of the FGT, which we further experimentally verified by current pulse width and temperature dependent experiments, as discussed below.

We investigated the dependence of the CIDWM on current density $J$, current pulse width $\tau$ and temperature $T$. Under fixed temperature and fixed pulse width, we first apply a low current density to drive the DW motion. Then the current intensity is gradually increased until a critical point where the domain wall structure is erased, where $v$ at this critical current density corresponds to its maximum value for this temperature and current pulse width i.e. $v_{\text{max}}$. This procedure is repeated for various pulse widths so as to obtain the pulse width dependence of $v_{\text{max}}$. The dependence of $v$ versus $J$ for different pulse widths at $T = 20$ K is shown in Fig. 1d. The functional form of the $v$ versus $J$ curves is similar, independent of pulse width, and follows the relationship $v_{\text{max}} \propto \sqrt{J^2 - \left(J_{th}^{\text{L}}\right)^2}$ [39]. Figure 1e shows $v_{\text{max}}$ as a function of pulse width $\tau$ at $T = 20$ K. The largest value of $v_{\text{max}}$ was found for the shortest $\tau$. As $\tau$ increases, $v_{\text{max}}$ initially drops quickly and then more slowly. The net heating power is given by the relation $(JA)^2R - p \propto \tau^{-1}$, where $A$ is the cross-sectional area of the device, $R$ is the device resistance, and $p$ represents the heat dissipation to both the substrate and the vacuum (top of the device) (see fit to the data in the inset to Fig. 1e). Thus, we found that the maximum value of $v_{\text{max}}$ corresponds to the lowest temperature and shortest pulse width, which is in agreement with the expectation that this corresponds to the heating of the FGT up to $T_c$ via the current.

These results confirm that the origin of $J_{th}^{\text{U}}$ derives from Joule heating: since the energy needed to heat the sample to $T_c$ is constant, for larger $\tau$ the temperature increase is greater during the duration of the pulse, so giving rise to a lower $J_{th}^{\text{U}}$; conversely, for shorter current pulses, a larger current density is required to reach $T_c$, thereby resulting in a higher velocity. Figure 1f shows the temperature dependence of the velocity for $\tau = 5$ ns. The velocity drops monotonically between 20 K and 140 K, and becomes zero when the temperature reaches $T_c \backsim 150$ K of the FGT flake.

Next, we consider the current-induced SOT switching of the magnetization of FGT thin flakes by fabricating heterostructures that consist of a heavy metal layer adjacent to the FGT. After exfoliation of FGT thin flakes in a N$_2$ glovebox, they were immediately transferred to a magnetron-sputtering deposition system via an ultra-high vacuum tube system, as shown in Fig. S3, without exposure to air, which guarantees a clean interface. A 3 nm thick platinum (Pt) or tungsten (W) layer is deposited onto the FGT flake to form a heavy metal/ferromagnetic metal heterostructure. As the current passes through the heavy metal layer, a vertical spin current is generated by the spin Hall effect and thus a SOT is exerted on the magnetization of the FGT thin flake. In the presence of a longitudinal magnetic field, $H_x = 300$ Oe, the magnetization of the FGT layer can be switched. Figure 2a shows differential Kerr microscope images of FGT/Pt heterostructure, formed by subtracting the prior Kerr image of the flake, so that the contrast corresponds to the change in the magnetization distribution. By reversing the applied magnetic field or current direction, the same current pulses switch the magnetization of the FGT into the opposite direction. When W is used instead of Pt, the switching behavior is reversed as expected from the known opposite signs of the spin Hall angles for Pt and W (Fig. 2b)[41,42]. These results indicate that the interface between the sputtered heavy metal layer and the FGT flake is sufficiently transparent to allow the spin current to flow across the interface. We note that full switching was not found when the FGT flake was exposed to air prior to deposition of the heavy metal layer, thereby showing the how easily the interface transparency between the FGT flake and the W or Pt can be damaged.

Following the successful demonstration of SOT current-induced switching of FGT, we explored CIDWM in FGT/Pt and FGT/W heterostructures. Clear evidence for CIDWM was found but the direction of motion of the DWs was found to depend on the magnitude of the current density for the FGT/Pt devices. For large enough current densities, the DW motion was found to be along the current direction rather than opposite to the current direction as was found for pure STT-driven motion, as discussed above (see Fig. 3a, b). However, for smaller current densities (see Fig. 3c, d), the DW moves in the direction opposite to the current direction. In both regimes, the velocities are -0.2–0.3 m/s, which is one order of magnitude slower than that in pristine FGT. Such a phenomenon can be well accounted for by the competition between SOT and STT-driven motion.

As distinct different from the FGT/Pt devices, the direction of the DW motion was found to be independent of the magnitude of $J$ for the FGT/W heterostructures. Indeed, the current-induced DW motion in FGT/W is in the same direction as that observed in pristine FGT and their velocities are similar, which shows that both the SOT and STT mechanisms drive the DWs in the same direction. However, an interesting difference between pristine FGT and FGT/W is the dependence of the DW velocity on an applied longitudinal magnetic field, $H_x$. For FGT/W, as shown in Fig. 4a, at $T = 20$ K, $v$ versus $H_x$ shows a slow decrease for positive $H_x$ for up/down DW configurations, and fast decreases for negative $H_x$. These are reversed for down/up DW configurations. In contrast, for pristine FGT devices, $v$ decreases with increasing $H_x$ for both negative and positive fields but the dome-like shape is offset from zero field with a peak around -+70 Oe (-−70 Oe) for up/down (down/up) DWs. These behaviors can be well described by the 1D CIDWM model that includes both STT and SOT[43]. The fact that the domes are offset to positive and negative fields for the

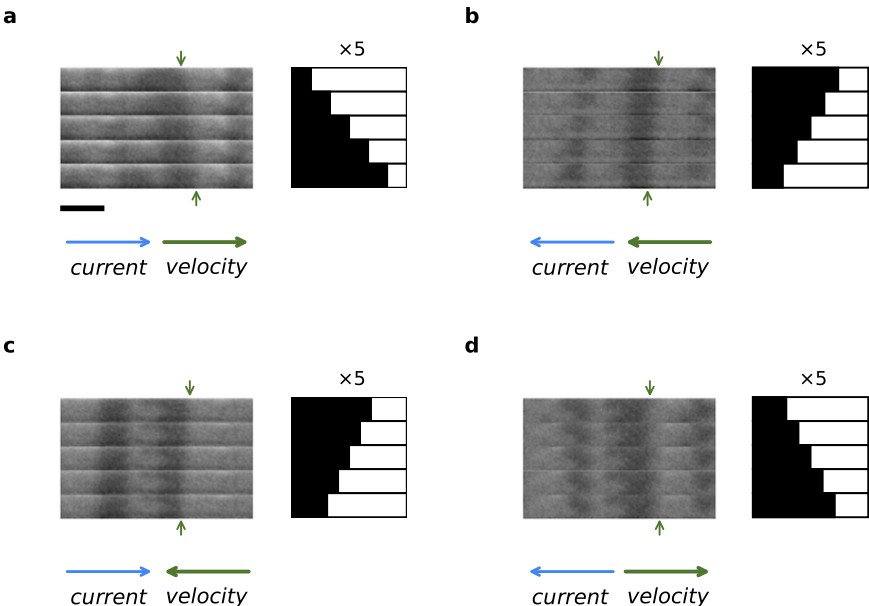

**Fig. 3 | Current-induced DW motion in a FGT (8.1 nm)/Pt(3 nm) heterostructure at $T = 70$ K.** Each differential Kerr image (from top to bottom) is taken after a series of 500 current pulses, each pulse is 2 ns long. The current direction is positive in **a** and **c** and is negative in **b** and **d**. The current density $J = 9.6 \times 10^{11}$ A m$^{-2}$ in **a** and **b** and $J = 9.4 \times 10^{11}$ A m$^{-2}$ in **c** and **d**. Green arrows at the top and bottom of each set of images indicate the initial and final position of the DW, respectively. The position of the DW has been magnified five times and displayed in a schematic sketch to the right of each set of images. The direction of DW motion is parallel to the direction of current in the high current density regime and is opposite in the low current density regime. All images are to the same scale: the scale bar in **a** is 2 μm.

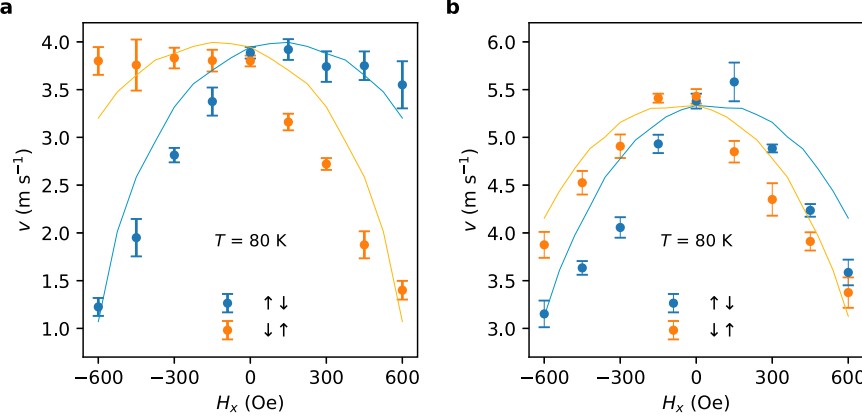

**Fig. 4 | Longitudinal field dependence of DW velocities in FGT/W hetero-structures and pristine FGT.** At $T = 80$ K, measured velocities in **a** FGT(9.1 nm)/W(3 nm) and **b** FGT(10.5 nm). Solid lines show the fitted DW velocities from a 1D model, as discussed in the main txt (the detailed fit parameters are shown in Table 1 in the SI). Blue and orange colors correspond to up/down and down/up DWs, respectively. Error bars are determined by multiplying the standard error of the mean by a factor of 1.96 to establish a 95% confidence interval.

up/down and down/up DWs in the pristine FGT flakes (Fig. 4b) demonstrates that there is an intrinsic bulk DMI so that the applied field adds or subtracts from the DMI effective exchange field which for Néel DWs is perpendicular to the DW. The even larger difference for the FGT/W case (Fig. 4a) is due to the additional SOT in this case. If the SOT were dominant then an applied longitudinal field comparable in magnitude and opposite to the DMI effective exchange field can reduce the DW velocity to zero and even reverse the direction when an even larger field is applied. Thus, the STT is the dominant driving torque here.

## Discussion

The current-induced DW velocities in our exfoliated flakes are an order of magnitude lower than those found in traditional ferromagnetic metallic magnetic materials, such as permalloy[44]. One reason is that the Curie temperature of FGT[12,13] is much lower than those of these others which, thereby, limits the maximum current density passing through FGT: this is an order of magnitude smaller than the maximum current density that can be used in permalloy nanowires[44]. If the Curie temperature can be enhanced[13], e.g., using a 2D compound with a higher Curie temperature[45,46], the DW velocity can be further improved. For the same reason, lowering the temperature of the device may allow for higher current densities, and thus DW velocities. However, since the specific heat of a material drops as the temperature decreases, Joule heating can readily warm up the device. Therefore, the DW velocity saturates at low temperatures, as shown in Fig. 1f. Another reason is the high spin-orbit coupling of the heavy Te atoms in FGT that will limit the spin-polarization of the current that is a result of spin-dependent scattering within the FGT. This is consistent with the high value of Gilbert damping reported in FGT[47].

The value of $J_{th}^L$ that originates from an intrinsic mechanism in the STT-case is proportional to the product of the transverse magnetic anisotropy energy $K_\perp$ and the DW width $\Delta$, according to $J_{th}^L = \frac{eM^2}{a^3\hbar}K_\perp\Delta$, where $e$ is the elementary charge, $M$ is the saturation magnetization, $a$ is the lattice constant, and $\hbar$ is the reduced Planck constant. This reflects the difference in energy between a Bloch type and a Néel type DW. With increasing pulse width, an increase in Joule heating can cause a decrease in $M$ and $K_\perp\Delta$ and thus a decrease of $J_{th}^L$. The intrinsic threshold current only appears when the non-adiabatic STT term is small. The suppression of the non-adiabatic term in the pristine FGT case is either due to a long spin-flip time (due to its metallic nature) or the large Gilbert damping parameter (~0.6)[47].

Our experimental results can be accounted for by a competition between SOT- and STT-driven DW motions for the FGT/Pt heterostructures. This is reflected by the lower DW speeds as compared to the case of pristine FGT, and by a change in the direction of the DW motion with increasing current density. When increasing the current density, the spin current from STT will decrease due to the decrease of saturation magnetization induced by Joule heating, while the spin current from SOT remains almost constant. The DW velocity in FGT/Pt can be divided into two components $\nu_{SOT}$ and $\nu_{STT}$, where the DW velocity induced by the STT, $\nu_{STT}$, is the same as that of pristine FGT for a specific current density. The DW velocity induced by the SOT, $\nu_{SOT}$ depends on the SHE of Pt layer and the DMI in FGT. In FGT/Pt with positive spin Hall angle of Pt, $\nu_{SOT}/\nu_{STT}$ is parallel/antiparallel to the current. As the current density increases, the transient temperature during the current pulse rises by Joule heating, the saturation magnetization and thus the spin current from STT are reduced whereas the spin current from SOT remains almost constant. Accordingly, the direction of DW motion changes with varying magnitude of the current. For FGT/W with a negative spin Hall angle of W, $\nu_{SOT}$ and $\nu_{STT}$ are both antiparallel to the current. Therefore, the direction of the DW motion is always antiparallel to that of the current and the CIDWM behavior is similar to that of the pristine FGT.

For the FGT/W case, instead of competing, the SOT and the STT assist each other in driving the DWs due to the opposite sign of the spin Hall angle in W as compared to that of Pt, which in turn gives rise to a CIDWM similar to that in pristine FGT. This is different from the SOT and DMI in conventional ferromagnetic metal racetrack devices, where when Pt is substituted by W, the SOT and DMI change their sign together. However, since in FGT, the DMI is of bulk origin rather than an interfacial origin, the DMI remains the same even when Pt is changed to W. The coexistence of the SOT- and STT-driven DW motion is confirmed by comparing the different longitudinal magnetic field dependencies of the DW velocity in FGT/W heterostructures and pristine FGT (Fig. 4). The different longitudinal field dependencies confirm the existence of an SOT from W layer and these dependences can be well accounted for within a 1D-DW model that includes both STT- and SOT-driven DW motion (see Table S1 for fitting parameters). It is worth noting that the DW velocity peaks at a non-zero longitudinal field in pristine FGT, which reflects a Néel type domain wall induced by the DMI[43]. In a system without DMI, both the velocity of the up/down and down/up DWs will peak at zero external field and the application of a longitudinal magnetic field will decrease the DW velocity (see Figure S3). The effective DMI field is found from the 1D-DW model fit to be ~70 Oe. Such a bulk DMI in pristine FGT flakes originates from both Fe vacancies and self-intercalated Fe atoms in the van der Waals gap[24].

In conclusion, we have studied CIDWM in the 2D ferromagnet FGT by magnetic optical Kerr microscopy imaging. CIDWM based on STT has been clearly displayed. The highest DW velocity of 5.68 m/s at 20 K is observed. The longitudinal magnetic field dependence of DW motion reveals a Néel type DW induced by DMI in pristine FGT. Pt and W film have been deposited on top of FGT to form heavy metal/

ferromagnet heterostructures. Opposite signs of spin Hall angles in Pt and W give rise to opposite SOT-induced magnetization switching directions and different behaviors of CIDWM. The competition between STT- and SOT-driven DW in FGT/Pt leads to a lower DW velocity and change of the DW motion direction with increasing current density while in FGT/W, the STT and SOT aid each other and leads to a DW motion as efficient as that in the pristine case. Such a DMI originates from the disorder of iron atom vacancies and intercalation of iron atoms in the van der Waals gap in the FGT flakes. Our work shed lights on the development of functional spintronic devices based on 2D magnets.

## Methods

### Device fabrication
FGT nanoflakes were directly exfoliated from a FGT signal crystal (HQ Graphene) onto a Si/285 nm SiO$_2$ wafer in a N$_2$ glovebox. A 950 K PMMA A5 e-beam resist was spin-coated onto the flakes as a protection layer. Standard e-beam lithography is used to pattern racetrack devices (Raith Pioneer 2 system). Ti (4 nm)/Au (40 nm) electrodes were deposited by magnetron-sputtering using a SCIA coating system. Ar ion-milling in the same SCIA coating system and focused ion beam milling in a TESCAN Ga-FIB GAIA system were used for etching away extraneous material.

### Kerr microscopy
A CryoVac cryostat was used to cool the racetrack devices using either liquid helium or liquid nitrogen as the refrigerant. Magnetic optical Kerr microscopy was employed to image the perpendicular magnetization of the device. A Nikon CFI S Plan Fluor ELWD 60XC object lens with numerical aperture of 0.7 was used.

## Data availability
The data that support the findings of this study are available from the corresponding author upon reasonable request.

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

## Acknowledgements

This project received funding from the Deutsche Fornschungsgemeinschaft (DFG, German Research Foundation)—project no. 403505322, Priority Programme (SPP) 2137. We also acknowledge support from the Samsung Electronics R&D program 'Material and Device Research on Racetrack Memory'. T.M. acknowledges support from the National Natural Science Foundation of China (no. 12304133).

## Author contributions

W.Z., T.M., and Y.G. performed the Kerr microscope measurements. W.Z., B.K.H., P.R.M.R., A.K.S., Z.W., K.G. fabricated the sample. H.M., P.R.M.R., Z.Y., and A.K.S. performed the structure measurements. Y.G. and S.H.Y. performed the 1D model calculation. T.M. built the current-induced domain wall motion measurement system. W.Z., T.M., Y.G., S.Z., S.W., and S.S.P.P. wrote the manuscript. W.Z., T.M., and S.S.P.P. conceived the project. All authors discussed the data and commented on the manuscript.

## Funding

## Competing interests

The authors declare no competing interests.
