## [Peer Review File · Nature Communications]

Current-induced domain wall motion in a van der Waals ferromagnet Fe₃GeTe₂REVIEWER COMMENTS

Reviewer #1 (Remarks to the Author):

In the manuscript “Current Induced domain wall motion in a van der Waals ferromagnet Fe₃GeTe₂” the authors present an experimental study of nanometer thick Fe₃GeTe₂ (FGT) flakes with and without Pt/W epilayers and they use current pulses to induce Domain Wall movement. Their main findings revolve around the competition between Spin Transfer Torque and Spin Orbit Torque effect in FGT/Pt and FGT/W heterostructures and their effect on domain mobility. Those findings are important for assessment of FGT based heterostructures as a viable candidate for novel spintronic devices.

General Comments:

The manuscript is well written in general. The authors could improve the Introduction section by stating more clearly the state of the art in FGT and the novelty of their approach. The arguments are presented in a clear and comprehensive way. The quality and novelty of those research findings are within the scope and status of the Journal. Despite that, there are a series of issues that need to be resolved before the publication of this manuscript in Nature Communications. I believe that in this way the clarity of the manuscript as well as its impact will be benefited.

Here I present enumerated the issues to be addressed:

1. In the Abstract (Line 20) the authors say: “Neel type skyrmions have been recently observed”, but in the Introduction of the manuscript a series of published papers regarding Neel type skyrmions in FGT are missing (For example 1) Mari lle J, et al, Nano Letters 2020 20 (12), 8563-8568. DOI: 10.1021/acs.nanolett.0c03111. 2) P Pappas et al 2023 2D Mater. 10 045033 DOI: 10.1088/2053-1583/acfb1f. 3) L. C. Peng, et al, Adv. Funct. Mater. 2021, 31, 2103583. DOI:10.1002/adfm.202103583. 4) Birch, M.T., Powalla, L., et al, Nat Commun 13, 3035 (2022). DOI:10.1038/s41467-022-30740-7, 5) Li, Y., et al, JOM 74, 2310–2318 (2022). DOI:10.1007/s11837-022-05299-9). The authors should explicitly comment on the research done on Neel type skyrmions in FGT by augmenting the Introduction with a respective paragraph.

2. The observed effects are attributed to bulk DMI due to vacancies and intercalation of iron atoms (lines 218-219). There are a series of published papers (As mentioned above 1) Mari lle J, et al, Nano Letters 2020 20 (12), 8563-8568. DOI: 10.1021/acs.nanolett.0c03111. 2) P Pappas et al 2023 2D Mater. 10 045033 DOI: 10.1088/2053-1583/acfb1f. 3) L. C. Peng, et al, Adv. Funct. Mater. 2021, 31, 2103583. DOI:10.1002/adfm.202103583, 5) Li, Y., et al, JOM 74, 2310–2318 (2022). DOI:10.1007/s11837-022-05299-9.) and also Ref [17], [18] in this manuscript attributing the emergence of similar phenomena to Interfacial DMI (from native oxide or the presence of an epilayer) or dipole-dipole interactions. The authors should comment on this research results in the Introduction. In Lines 147-150, the authors mainly support the presence of bulk DMI in the offset seen in Fig. 4, but this argument is a little bit vague since other effects can result in such offset. The authors should further support that there is indeed bulk DMI and the associated structural defects in their samples. Maybe the presence of an effective exchange field due to DMI can be presented by MOKE magnetometry hysteresis loops.

3. In Lines 203-204 the authors suggest that the domain walls are of Neel type based on the

dependence of domain wall velocity to the external magnetic field. The authors should elaborate this argument more.

4. Lines 128-142 the Figure references are not clear. It is stated that Fig. 3b corresponds to smaller current densities but Fig. 3 caption associates the higher current densities to Fig. 3a and b and lower current densities to Fig. 3c and d. The authors should clarify the content of this paragraph. Furthermore Fig. 3c and 3d are not clearly mentioned in the manuscript.

5. The authors compare the current driven domain wall motion of pristine FGT and FGT heterostructures but the thickness of the FGT flakes is not the same. It varies from in 4.1 nm in the case of FGT/Pt to 8.6 nm in FGT/W and 10.5 nm in pristine FGT. The authors should comment in the manuscript why the variation of FGT thickness does not affect the results presented.

6. The authors state that their flakes have a T_c around 150 K. Based on the literature (for example Ref 12 in this manuscript) this value is a little bit low, especially for the flakes with 10 nm thickness where a T_c close to 190 K is expected. Have the authors verified the T_c by MOKE magnetometry for all flakes? If yes this can be commented in the manuscript and a representative MOKE contrast vs T figure added in the Supplementary material section. Can this low T_c be associated with the defects the authors claim to cause the emergence of bulk DMI? If so, the authors should clearly state it in the manuscript.

7. The domain wall velocity is calculated from the MOKE images as shown in Fig. 1 at 20 K. The domain wall velocities presented in In Fig. 4 is at 80 K, the authors should provide the corresponding MOKE images at 80 K in the Supplementary Information section.

8. In Lines 233-235 the authors should add the specs of the lens (magnification, numerical aperture) used during MOKE experiments.

Reviewer #2 (Remarks to the Author):

The work explores current-induced domain wall motion in thin (<10 nm) exfoliated flakes of FGT, both pristine and in heterostructures with both Pt and W, via direct imaging of the domains. Previous works have either looked at much thicker flakes or used indirect transport measurements. The dependence of the domain wall velocity and direction on current density, pulse width, temperature, and longitudinal magnetic field, is shown to be well explained by the presence of STT in the pristine flakes added to SOT from the metallic layer when present (with opposite signs for Pt and W) acting on Neel domain walls stabilised by bulk DMI. This interpretation is convincing and very interesting, and confirms previous works while providing more direct evidence. Data is reasonably well presented and the paper easy to follow. The work is of high quality and would be interesting to those in the 2D van der Waals magnet community. I recommend publication in Nat Comms after the authors addressed the following comments:

1. What is the thickness of the FGT flake in Fig. 1? It would be interesting to study the dependence on thickness for the pristine flake experiments.

2. FGT was shown to host an intrinsic SOT effect [doi:10.1002/adma.202004110, doi:10.1080/21663831.2022.2119108]. The authors should comment on why they believe

STT is responsible for the current induced domain wall motion in the pristine FGT flakes and not the intrinsic SOT.

4. The contrast in the MOKE images varies significantly between figures 1, 2, and 3. Can this observation be explained? This is particularly relevant in Fig. 3d where the change in domain wall motion is significantly less obvious than in the rest of the presented images.

5. Domain nucleation is a necessary aspect of studying domain wall motion. The authors should comment on how they initialised their devices before current injection.

6. In the introduction, the authors state that direct observation of CIDWM in a few-layer thick FGT is currently lacking. It may be worth citing a work done on thin FGT flakes using NV-diamond imaging, though their study of CIDWM was very rudimentary: doi:10.1088/2053-1583/acab73

Reviewer #1 (Remarks to the Author):

In the manuscript “Current Induced domain wall motion in a van der Waals ferromagnet Fe₃GeTe₂” the authors present an experimental study of nanometer thick Fe₃GeTe₂ (FGT) flakes with and without Pt/W epilayers and they use current pulses to induce Domain Wall movement. Their main findings revolve around the competition between Spin Transfer Torque and Spin Orbit Torque effect in FGT/Pt and FGT/W heterostructures and their effect on domain mobility. Those findings are important for assessment of FGT based heterostructures as a viable candidate for novel spintronic devices.

General Comments:

The manuscript is well written in general. The authors could improve the Introduction section by stating more clearly the state of the art in FGT and the novelty of their approach. The arguments are presented in a clear and comprehensive way. The quality and novelty of those research findings are within the scope and status of the Journal. Despite that, there are a series of issues that need to be resolved before the publication of this manuscript in Nature Communications. I believe that in this way the clarity of the manuscript as well as its impact will be benefited.

We thank the reviewer for his/her very positive comments about our manuscript.

Here I present enumerated the issues to be addressed:

1. In the Abstract (Line 20) the authors say: “Neel type skyrmions have been recently observed”, but in the Introduction of the manuscript a series of published papers regarding Neel type skyrmions in FGT are missing (For example 1) Mari lle J, et al, Nano Letters 2020 20 (12), 8563-8568. DOI: 10.1021/acs.nanolett.0c03111. 2) P Pappas et al 2023 2D Mater. 10 045033 DOI: 10.1088/2053-1583/acfb1f. 3) L. C. Peng, et al, Adv. Funct. Mater. 2021, 31, 2103583. DOI:10.1002/adfm.202103583. 4) Birch, M.T., Powalla, L., et al, Nat Commun 13, 3035 (2022). DOI:10.1038/s41467-022-30740-7, 5) Li, Y., et al, JOM 74, 2310–2318 (2022). DOI:10.1007/s11837-022-05299-9). The authors should explicitly comment on the research done on N el type skyrmions in FGT by augmenting the Introduction with a respective paragraph.

We thank the reviewer for sharing these papers. We have added a paragraph to the introduction, which includes these papers as well as other related articles, and we discuss the underlying mechanism for the formation of N el type skyrmions in FGT.

2. The observed effects are attributed to bulk DMI due to vacancies and intercalation of iron atoms (lines 218-219). There are a series of published papers (As mentioned above 1) Mari lle J, et al, Nano Letters 2020 20 (12), 8563-8568. DOI: 10.1021/acs.nanolett.0c03111. 2) P Pappas et al 2023 2D Mater. 10 045033 DOI: 10.1088/2053-1583/acfb1f. 3) L. C. Peng, et al, Adv. Funct. Mater. 2021, 31, 2103583. DOI:10.1002/adfm.202103583, 5) Li, Y., et al, JOM 74, 2310–2318 (2022). DOI:10.1007/s11837-022-05299-9.) and also Ref [17], [18] in this manuscript attributing the emergence of similar phenomena to Interfacial DMI (from native oxide or the presence of an epilayer) or dipole-dipole interactions. The authors should comment on this research results in the Introduction. In Lines 147-150, the authors mainly support the presence of bulk DMI in the offset seen in Fig. 4, but this argument is a little bit vague since other effects can result in such offset. The authors should further support that there is indeed bulk DMI and the associated structural defects in their samples. Maybe the presence of an effective exchange field due to DMI can be presented by MOKE magnetometry hysteresis loops.

We thank the reviewer for these comments. We discuss the origin of the DMI in detail in the paragraph that we have added to the introduction. We are convinced that the skyrmions in FGT arise from a bulk mechanism for the following reasons:

- a. In our previous study [1], through careful structure determination and analysis, we show that the FGT structure of a single crystal, from which the flakes are prepared, possesses inversion symmetry breaking. This is due to different occupancies of certain Fe Wykoff sites (from Fe vacancies) which leads to the symmetry lowering. This leads to a C_{3v} point group symmetry rather than the D_{6h} point group symmetry that was long believed to be that of the FGT structure. Whereas the C_{3v} point group supports the stabilization of Néel skyrmions the D_{6h} point group does not. This is why several groups proposed an interfacial origin of the DMI that is caused e.g. by oxidation of the surfaces of the FGT flakes. However, such an interfacial mechanism would seem to be too weak to stabilize skyrmions in the thick layers that were reported in these other papers. Note that the determination of the lowered symmetry of the structure from XRD is non-trivial since only a tiny proportion of the atoms in the unit cell are involved. Note that in FGT the Fe component is only 50% of the total number of atoms and the Fe vacancies are an even lower proportion (a few percent). The corresponds to only 8 electrons out of 428 electrons in the unit cell. This subtle, but very important bulk structural modification had previously been overlooked due to the difficulty in determining it.
- b. In our recent work [2], we confirmed, via detailed XRD analysis, that the structure of an individual flake, only 22 nm thick, also has the same lowered crystal symmetry as the single crystal.
- c. Recently, in an independent study [3], another group has published support for our C_{3v} FGT crystal structure. In this work, this group uses the second harmonic generation (SHG) technique to determine the symmetry of the crystal structure of FGT. Their results confirm the C_{3v} point group symmetry and thus provide support for our bulk DMI model.
- d. As we had discussed in our original manuscript the dependence of the current induced domain wall motion (CIDWM) velocity in the presence of an applied longitudinal magnetic field supports the Néel type configuration of the domain walls in FGT flakes. This external field adds or subtracts from the DMI field that is perpendicular to the domain walls in such walls. It is difficult to extract the DMI field from M-H loops that the reviewer suggests. This is because the magnetization is either a single domain state, or there are equal numbers of up/down and down/up domain walls.

3. In Lines 203-204 the authors suggest that the domain walls are of Néel type based on the dependence of domain wall velocity to the external magnetic field. The authors should elaborate this argument more.

In our work, the domain wall motion is driven by STT and shows a dome-like dependence on the longitudinal (x) field (see Fig. R1). We use the analytical 1D model, which is widely used, to analyze the domain wall dynamics [4]. We conclude from fitting to this 1D model that the DWs in FGT are Néel type since the dome peaks at a non-zero x -field: moreover, as driven by adiabatic torque from STT, the DW motion is a precessional mode: the DW oscillates between Bloch and Néel type configurations when it moves and accordingly the DW's velocity is modulated. The DW reaches its maximum speed when the energy of the Bloch type and Néel type DW configurations are equal to each other, so that the DW attains a peak velocity when the DMI effective field is compensated by a x -field of the opposite sign. From the 1D analytical model, if we do not have any DMI effective field, the longitudinal field dependence is the same for Up/Down and Down/Up DWs so that the DW velocity peaks at zero field. This is in sharp contrast with our FGT data where the velocity of the Up/Down and Down/Up DWs peak at non-zero values which are of opposite sign. From this point of view, we conclude that we have a Néel type originating from the DMI in our FGT. We have added the related discussion in the revised manuscript. Fig. R1 is also added to supporting information as Figure S3.

Fig. R1 Longitudinal field dependence of DW velocities from simulation based on 1D analytical model. a, with DMI and b, without DMI.

4. Lines 128-142 the Figure references are not clear. It is stated that Fig. 3b corresponds to smaller current densities but Fig. 3 caption associates the higher current densities to Fig. 3a and b and lower current densities to Fig. 3c and d. The authors should clarify the content of this paragraph. Furthermore Fig. 3c and 3d are not clearly mentioned in the manuscript.

We thank the reviewer for pointing this out. Both Fig. 3a and Fig. 3b correspond to smaller current densities regime but with different directions of current, while Fig. 3c and Fig. 3d correspond to higher current densities regimes. This has been corrected in the revised main text as follows: “for pure STT driven motion, as discussed above (see Fig. 3a and 3b). However, for smaller current densities (see Fig. 3c and 3d) the DW moves in the direction opposite to the current direction.”

5. The authors compare the current driven domain wall motion of pristine FGT and FGT heterostructures but the thickness of the FGT flakes is not the same. It varies from in 4.1 nm in the case of FGT/Pt to 8.6 nm in FGT/W and 10.5 nm in pristine FGT. The authors should comment in the manuscript why the variation of FGT thickness does not affect the results presented.

We thank the reviewer for this comment. Due to the mechanical exfoliation technique, it is difficult to prepare FGT flakes with a series of chosen thickness. As shown in Fig. R2, orange circles represent the data from Fig. 1f in the main text for a flake which is 10.5 nm thick, while blue circles represent the data from an FGT flake of 9 nm thick. Though their T_c are slightly different due to their different thicknesses, their temperature dependences are similar. Fig. R2 is inserted in supporting information as Figure S4.

For switching in heterostructures, another pair of FGT(6.7 nm)/Pt(3 nm) and FGT(8.6 nm)/W(3 nm) with closer thicknesses than the pair in the main text are demonstrated in Fig. R3. They behave similarly to those ones present in the main text.

Fig. R2 Thickness dependence of velocity of CIDWM in pristine FGT. Orange curve shows the velocity in a sample with thickness of 10.5 nm. Blue curve shows the velocity in another sample with thickness of 9 nm. Due to the different T_c of two samples, curves are normalized as $T_c - T$ versus v_{\max} .

Fig. R3 Current induced spin-orbit torque switching in $\text{Fe}_3\text{GeTe}_2/\text{Pt}$ and $\text{Fe}_3\text{GeTe}_2/\text{W}$ heterostructure. Differential Kerr microscope images of **a**, $\text{Fe}_3\text{GeTe}_2(6.7 \text{ nm})/\text{Pt}(3 \text{ nm})$ **b**, $\text{Fe}_3\text{GeTe}_2(8.6 \text{ nm})/\text{W}(3 \text{ nm})$. Black and white represents magnetization up and down, respectively. The series of current pulses passes through the heterostructure along x -axis, while the applied longitudinal field $H = 300 \text{ Oe}$ is also parallel to the x -axis. The scale bars are $5 \mu\text{m}$.

6. The authors state that their flakes have a T_c around 150 K. Based on the literature (for example Ref 12 in this manuscript) this value is a little bit low, especially for the flakes with 10 nm thickness where a T_c close to 190 K is expected. Have the authors verified the T_c by MOKE magnetometry for all flakes? If yes this can be commented in the manuscript and a representative MOKE contrast vs T figure added in the Supplementary material section. Can this low T_c be associated with the defects the authors claim to cause the emergence of bulk DMI? If so, the authors should clearly state it in the manuscript.

We thank the reviewer for making this point. We determined the T_c of our flakes using MOKE: most of them fall in the range of 150 - 175 K. Fig. R4 shows the temperature dependence of the MOKE contrast of a 9 nm FGT flake with $T_c = 172 \text{ K}$, which is close to the value reported (Fig. 2d in [5]). There is a discrepancy between T_c measured by different methods, due to the existence of small domains

with higher T_c than average. This may also explain our results, owing to the resolution of our Kerr microscope. We believe therefore that our results are similar to those reported. We have added Fig. R4 into supporting information as Figure S5.

Fig. R4 Temperature dependence of the MOKE contrast of a 9 nm FGT flake. Data are fitted by the critical power-law form $(1-T/T_c)^\beta$, where T_c is 172 K and β is 0.25[6].

7. The domain wall velocity is calculated from the MOKE images as shown in Fig. 1 at 20 K. The domain wall velocities presented in In Fig. 4 is at 80 K, the authors should provide the corresponding MOKE images at 80 K in the Supplementary Information section.

We thank the reviewer for this comment. In Fig. R5, corresponding MOKE images at 80 K are demonstrated. We have added these to the supporting information (as Figure S6), as the reviewer requests.

Fig. R5 MOKE images of the CIDWM under various longitudinal fields of a 9 nm FGT flake at 80 K.

8. In Lines 233-235 the authors should add the specs of the lens (magnification, numerical aperture) used during MOKE experiments.

We thank the reviewer for this suggestion. We used a Nikon CFI S Plan Fluor ELWD 60XC lens, which has a magnification of 60X and NA of 0.7. We have added this to the revised manuscript.

Reviewer #2 (Remarks to the Author):

The work explores current-induced domain wall motion in thin (<10 nm) exfoliated flakes of FGT, both pristine and in heterostructures with both Pt and W, via direct imaging of the domains. Previous works have either looked at much thicker flakes or used indirect transport measurements. The dependence of the domain wall velocity and direction on current density, pulse width, temperature, and longitudinal magnetic field, is shown to be well explained by the presence of STT in the pristine flakes added to SOT from the metallic layer when present (with opposite signs for Pt and W) acting on Neel domain walls stabilised by bulk DMI. This interpretation is convincing and very interesting, and confirms previous works while providing more direct evidence. Data is reasonably well presented and the paper easy to follow. The work is of high quality and would be interesting to those in the 2D van der Waals magnet community. I recommend publication in Nat Comms after the authors addressed the following comments:

We thank the reviewer for his/her very positive comments about our manuscript. A point-by-point response is presented below and the manuscript is modified accordingly.

1. What is the thickness of the FGT flake in Fig. 1? It would be interesting to study the dependence on thickness for the pristine flake experiments.

We thank the reviewer for carefully read our manuscript and his/her valuable suggestions. The thickness of the FGT flake in Fig. 1 is 10.5 nm, we have added this to the revised manuscript. In Fig. R2, we show the temperature dependence of the maximum domain wall velocity of pristine FGT with two different thicknesses: 10.5 nm as in the main text and 9 nm. Though their T_c are slightly different (around 20 K) due to the thickness difference, their temperature dependence are almost identical to each other. We agree that further investigations on the thickness dependence of the CIDWM is highly interesting but somehow itself can be a both challenging and independent work.

2. FGT was shown to host an intrinsic SOT effect [doi:10.1002/adma.202004110, doi:10.1080/21663831.2022.2119108]. The authors should comment on why they believe STT is responsible for the current induced domain wall motion in the pristine FGT flakes and not the intrinsic SOT.

We thank the reviewer for this comment. Intrinsic SOT in FGT is an interesting new mechanism to be further explored, while in this manuscript we believe STT is the dominating effect in the CIDWM of our pristine FGT for the following reasons:

- a. As a ferromagnetic material, STT exists in the FGT system and should plays an important role in the current-induced magnetic dynamics. In our work, for pristine FGT devices, the DW moves along the electron flow direction, which is a special feature of STT-driven DW motion. In the previous work[7] for FGT motion in devices prepared by Focused-Ion-Beam, the current induced domain wall motion is also attributed to the STT.
- b. In the two works addressed by the reviewer, the SOT corresponds to an effective field of 50 Oe when the current density is 1 mA/um², which is 1-2 orders larger than that of Pt. If the intrinsic SOT contribution in the same way as SOT injected by Pt, the CIDWM velocity should be even higher than that of Pt/CoNiCo system above 100 m/s, which is not the case in our measurement. On the contrary, reference [8] concludes that the intrinsic SOT is act as an effective anisotropy. According to Eq. (6) within the reference, by applying the current, the perpendicular magnetic anisotropy (PMA) of the FGT becomes smaller. Since the change of anisotropy won't influence the CIDWM direction, we conclude the dominating torque that make the domain wall move is still STT.

- c. In reference [8] mentioned by the reviewer, only field-induced magnetization switching is studied in the presence of current. In reference [9] the intrinsic SOT is measured using a transport technique. How any such intrinsic SOT will influence the current-induced motion of domain walls is interesting but not yet understood.
- d. Recently, another work [3] has been published by the same group as in Ref [7]. In this new work, they connect the intrinsic SOT of FGT with the inversion symmetry broken crystal structure of FGT (C_{3v}) that we had proposed. However, in the first (theory) paper where the intrinsic SOT of FGT was proposed [10], the theoretical model was based on structures with D_{6h} or D_{3h} symmetry. Thus, the origin of the intrinsic torque in FGT remains to be explored and is not possible for us to discuss in our paper.

We added the corresponding modification in the results part of the main manuscript. We hope the reviewer find our comment and modification convincing.

4. The contrast in the MOKE images varies significantly between figures 1, 2, and 3. Can this observation be explained? This is particularly relevant in Fig. 3d where the change in domain wall motion is significantly less obvious than in the rest of the presented images.

We thank the reviewer for his/her carefully examination on our experiment. The Kerr contrast of the exfoliated FGT layer indeed varies between devices, mainly due to the following two reasons:

- a. With Pt or W capping layer, the absorption of light will be quite different so that the contrast will be different.
- b. There is an interference effect for 2D materials. For example, in [11] in $\text{Cr}_2\text{Ge}_2\text{Te}_6$ system, it is reported that when the sample thickness changes from 22.2 nm to 35.7 nm for the same light source, or when the light source wavelength changes from 532nm to 633nm for the same thickness, the Kerr contrast changes and its sign can even change.

So, for our racetrack device with different thicknesses and top metal layers, a significant change in contrast is expected. Fortunately, these contrast changes won't influence the DW position determination and the CIDWM measurements.

5. Domain nucleation is a necessary aspect of studying domain wall motion. The authors should comment on how they initialised their devices before current injection.

The method to initialize our devices is as follows. First, a large out-of-plane field 1100 Oe is applied and then removed to generate a single domain state in the FGT flakes. Next, a current pulse with a density above J_{th}^{U} is applied to create multi-domains in the device by joule heating. Finally, multiple pulses with current densities between J_{th}^{L} and J_{th}^{U} along both directions are applied consecutively to merge the domain walls until only one DW remains.

6. In the introduction, the authors state that direct observation of CIDWM in a few-layer thick FGT is currently lacking. It may be worth citing a work done on thin FGT flakes using NV-diamond imaging, though their study of CIDWM was very rudimentary: doi:10.1088/2053-1583/acab73

We appreciate the reviewer mentioning this paper and agree with the reviewer that this work is very rudimentary. Thus, we think the relevance of this literature to our work is high enough to be cited.

1. Chakraborty, A., et al., *Magnetic Skyrmions in a Thickness Tunable 2D Ferromagnet from a Defect Driven Dzyaloshinskii–Moriya Interaction*. *Advanced Materials*, 2022. **34**(11): p. 2108637.
2. Pal, B., et al. *Realization of a Spin Glass in a two-dimensional van der Waals material*. 2024. arXiv:2403.02088.
3. Zhang, K.-X., et al., *Broken Inversion Symmetry in Van Der Waals Topological Ferromagnetic Metal Iron Germanium Telluride*. *Advanced Materials*, 2023. **n/a**(n/a): p. 2312824.
4. Yang, S.-H., K.-S. Ryu, and S. Parkin, *Domain-wall velocities of up to 750 m s⁻¹ driven by exchange-coupling torque in synthetic antiferromagnets*. *Nature Nanotechnology*, 2015. **10**(3): p. 221-226.
5. Deng, Y., et al., *Gate-tunable room-temperature ferromagnetism in two-dimensional Fe₃GeTe₂*. *Nature*, 2018. **563**(7729): p. 94-99.
6. Fei, Z., et al., *Two-dimensional itinerant ferromagnetism in atomically thin Fe₃GeTe₂*. *Nature Materials*, 2018. **17**(9): p. 778-782.
7. Yang, C., et al., *Elevation of Domain Wall Velocity Driven by Current Pulses in 2D Ferromagnetic Material Fe₃GeTe₂*. *Advanced Functional Materials*, 2022. **32**(41): p. 2205144.
8. Zhang, K., et al., *Gigantic current control of coercive field and magnetic memory based on nanometer-thin ferromagnetic van der Waals Fe₃GeTe₂*. *Advanced Materials*, 2021. **33**(4): p. 2004110.
9. Martin, F., et al., *Strong bulk spin–orbit torques quantified in the van der Waals ferromagnet Fe₃GeTe₂*. *Materials Research Letters*, 2023. **11**(1): p. 84-89.
10. Johansen, Ø., et al., *Current control of magnetism in two-dimensional Fe₃GeTe₂*. *Physical Review Letters*, 2019. **122**(21): p. 217203.
11. Ma, Z., et al., *Micro-MOKE with optical interference in the study of 2D Cr₂Ge₂Te₆ nanoflake based magnetic heterostructures*. *AIP Advances*, 2019. **9**(12).

REVIEWERS' COMMENTS

Reviewer #1 (Remarks to the Author):

I would like to thank the authors for the time and effort they dedicated for making the changes suggested. I feel that all the question raised are adequately addressed by the authors. In this sense I support the publication of the manuscript to Nature Communications

Reviewer #2 (Remarks to the Author):

I am satisfied with the authors' response and revisions. I recommend publication in Nat Comms.